

# Beyond Observed Extremes: Can Hybrid Deep Learning Models Improve Flood Prediction?

Xiaoxiang Guan[1], Baoying Shan[2], Viet Dung Nguyen[1], Bruno Merz[1,3]

[1]GFZ Helmholtz Centre for Geosciences, Section Hydrology, Potsdam, 14473, Germany
[2]Dipartimento di Ingegneria Civile e Ambientale (DICA), Politecnico di Milano, Milan, 20133, Italy
[3]Institute for Environmental Sciences and Geography, University of Potsdam, Potsdam, 14476, Germany

*Correspondence to*: Xiaoxiang Guan (guan@gfz.de)

**Abstract.** Predicting unprecedented floods is essential for disaster risk reduction and climate adaptation but remains a challenge for both hydrological and deep learning models. This study evaluates three hydrological models, a Long Short-
Term Memory (LSTM) network, and three hybrid models in simulating extreme floods in more than 400 catchments in Central Europe. The hybrid models integrate hydrological process variables with meteorological inputs to enhance runoff simulations. Results show that the LSTM model outperforms traditional hydrological models, while hybrid models further reduce runoff simulation errors. However, all models tend to underestimate peak discharges, with over 50% underestimation for unprecedented floods. LSTM-based models exhibit extrapolation limits, likely due to structural and statistical constraints.
To improve extrapolation to rare events, future work should integrate physical principles into deep learning, including differentiable hydrological models, physics-guided loss functions, and synthetic extreme event generation. Additionally, regional modeling approaches, such as entity-aware LSTMs, could improve predictions by leveraging spatial hydrological similarities. Combining data-driven learning with physical reasoning will be key to improving flood simulations beyond observed extremes.

**1 Introduction**

Floods are among the most devastating natural hazards, causing significant economic losses, environmental degradation, and loss of life worldwide (Kreibich et al., 2022; Merz et al., 2021; Winsemius et al., 2016). Climate change, intensified land use and economic and population growth are contributing to an increase in the frequency and losses of extreme flood events, making effective flood prediction critical for water resource management and disaster preparedness (Alfieri et al., 2017;
Blöschl et al., 2015; Sun et al., 2024; Tarasova et al., 2023). In particular, unprecedented flood events—characterized by extreme discharges beyond the historical record (Heinrich et al., 2024)—pose unique challenges to modeling, prediction and management (Diffenbaugh et al., 2017; Kreibich et al., 2022; Merz et al., 2015).

Traditional physics-based hydrological models, while fundamental to flood prediction, struggle to accurately simulate such extreme events (Chen et al., 2021; Ma et al., 2024; Xu et al., 2024), especially when the floods fall outside the range of
observed data. The traditional hydrological models, which are built on predefined physical relationships and calibrated using



historical data, are limited in their ability to capture the complex and nonlinear interactions that often drive extreme flood events (van Kempen et al., 2021). This limitation becomes particularly apparent for unprecedented events, which are often characterized by conditions—such as extreme precipitation patterns or unusual catchment responses—that deviate significantly from those observed historically. Traditional models tend to underestimate peak discharges under these

conditions, as they are inherently constrained by their reliance on past data and predefined model structures (Adera et al., 2024). These limitations underscore the need for models that are more adaptive and more capable of extrapolation; consequently, there is a growing interest in advanced data-driven techniques, such as deep learning, to improve flood prediction capabilities under extreme or unprecedented conditions (Ma et al., 2024; Xu et al., 2024).

Advances in deep learning offer new opportunities to improve flood modeling by leveraging large datasets and identifying

complex, nonlinear patterns within them. Deep learning models, such as Long Short-Term Memory (LSTM) networks, have shown substantial promise in hydrological applications due to their ability to capture temporal dependencies and adapt to complex, dynamic environmental data (Adera et al., 2024; Chen et al., 2021; Deng et al., 2024; Koch and Schneider, 2022; Kratzert et al., 2018; Panahi et al., 2023; Rasheed et al., 2022). LSTM networks can simulate hydrological processes without the need for extensive calibration of physical parameters, potentially offering advantages in situations where traditional

models fall short (Koch and Schneider, 2022; Kratzert et al., 2018; Liu et al., 2024). However, purely data-driven models face limitations when applied to unprecedented flood events, as they may lack physical insights and may not effectively generalize to conditions beyond their training data (Adera et al., 2024).

Hybrid modeling approaches, which integrate the strengths of traditional physics-based hydrological models with deep learning techniques (Willard et al., 2022), have emerged as a promising solution for flood prediction (Adera et al., 2024;

Kraft et al., 2022; Li et al., 2023; Liu et al., 2024). Hybrid models aim to leverage the process-based understanding embedded in hydrological models, such as rainfall-runoff relationships, and enhance it with the data-driven adaptability of deep learning (Karniadakis et al., 2021). By combining these approaches, hybrid models may have the potential to capture both the underlying physical processes and the temporal dynamics associated with extreme meteorological conditions, potentially improving predictive performance for both typical and unprecedented floods.

Despite promising preliminary findings, further investigation is needed to assess whether a hybrid modeling strategy can accurately capture flood events, particularly very extreme scenarios not seen during training. Previous evaluations have relied on metrics such as the Nash Sutcliffe Efficiency (NSE), Kling-Gupta Efficiency (KGE) and high flow bias (FHV), none of which are designed to diagnose floods with unseen magnitudes (Frame et al., 2022; Lin et al., 2021; Xie et al., 2021; Xu et al., 2024). In particular, the role of physics-based hydrological model structures within hybrid modeling approaches

remains largely unexplored. It is unclear whether inadequate model structures lead to inaccurate representations of flood peaks or if incorporating hydrological process knowledge could improve predictions of unprecedented events. Furthermore, unprecedented floods vary widely in magnitude; events with magnitudes closer to historical floods may be simulated with greater accuracy than those of extraordinary scale. Further research is needed to assess how flood magnitude impacts model performance, particularly for extreme cases.





Here, we evaluate three physics-based hydrological models with varying model structures to assess their effectiveness in simulating the rainfall-runoff process and capturing unprecedented floods across more than 400 catchments in Central Europe. Their performance is compared to a purely data-driven Long Short-Term Memory (LSTM) model and three hybrid models, each integrating an LSTM with one of the hydrological models. This comparison aims to determine whether the hybrid approach improves the prediction of peak discharges during extreme events. Additionally, we explore the effects of model structure and flood magnitude on the accuracy of peak discharge representation for unprecedented floods.

## 2 Study area and data

Central Europe is selected as study area, with data sourced from the LamaH-CE dataset (LArge-SaMple DAta for Hydrology and Environmental Sciences for Central Europe, Klingler et al., 2021). This dataset aligns with the trend toward large-sample hydrology datasets designed to analyze hydrological processes across diverse and extensive regions (do Nascimento et al., 2024; Kratzert et al., 2023). LamaH covers approximately 170 000 km² across nine countries, ranging from lowland regions with a continental climate to high alpine zones dominated by snow and ice. For this study, a total of 416 river catchments with minimal or negligible anthropogenic influence were selected to assess flood modeling performance (Figure 1). Key input variables—including catchment-scale daily average precipitation, air temperature (average and maximum), and potential evapotranspiration—are derived from the LamaH-CE dataset. These inputs drive the physics-based and data-driven models to simulate runoff. The historical observation period spans from 1981 to 2017 for most catchments, providing a comprehensive dataset for model evaluation.

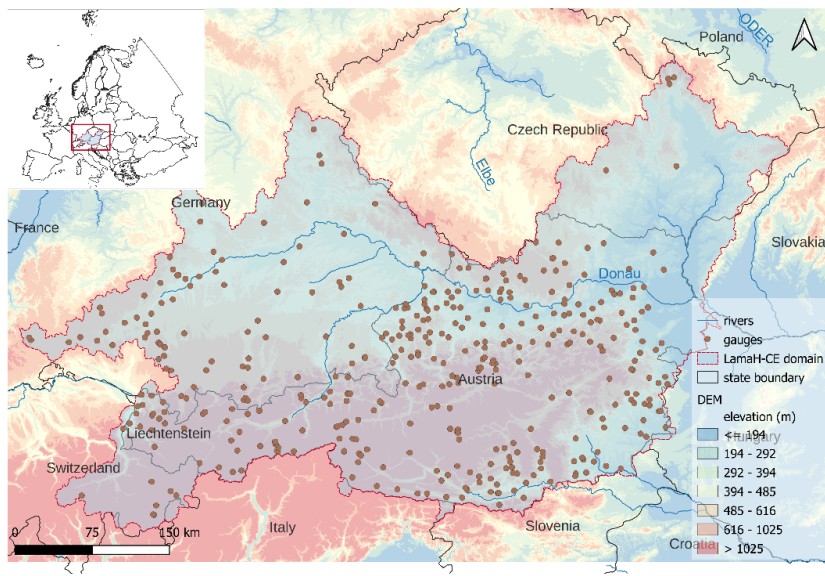

**Figure 1. Study area in Central Europe and the 416 selected gauges.**



# 3 Methods

## 3.1 Hydrological models

Three physics-based, lumped and continuous hydrological models (HBV, SIMHYD, and GR4J) are applied to evaluate the impact of different model structures on the modeling performance of unprecedented floods. These models are selected due to their widespread use in diverse hydrological conditions and their simplicity and low parameter requirements.

The HBV (Hydrologiska Byråns Vattenbalansavdelning) model has been widely used for rainfall-runoff simulation (Acuña

Espinoza et al., 2024; Macdonald et al., 2024, Mendez and Calvo-Valverde, 2016; Seibert and Vis, 2012). Using daily or hourly time steps, the model is forced by precipitation, air temperature, and monthly estimates of potential evapotranspiration. The snow routine calculates snow accumulation and melt via a degree-day approach. The soil routine simulates groundwater recharge, water storage and actual evaporation based on soil moisture. In the response routine, runoff is modeled as a function of water storage, while the routing routine employs a triangular weighting function to calculate

runoff at the catchment outlet (For details see Seibert and Vis, 2012).

The SIMHYD model has been applied successfully in many semi-arid or semi-humid basins, for instance in the United States and Australia (Chiew et al., 2002; Guan et al., 2019; Kachroo, 1992). SIMHYD simulates runoff through three components: surface flow, interflow, and base flow. Surface flow is generated by infiltration excess runoff when rainfall exceeds the infiltration capacity. Interflow occurs via saturation excess, representing subsurface flow when the soil is fully

saturated. Base flow is simulated as a linear recession of groundwater storage, sustaining streamflow during dry periods. Infiltration is modeled as a negative exponential function of soil moisture, dynamically adjusting runoff based on soil wetness (For details see Chiew et al., 2002).

The GR4J (Génie Rural à 4 paramètres Journalier) model is a rainfall-runoff model, designed to balance simplicity with robust performance (Edijatno et al., 1999). GR4J has gained widespread popularity in hydrological research and operational

water management, being applied in diverse hydrological contexts across many countries worldwide (Guan et al., 2020; Kuana et al., 2024; Perrin et al., 2003; Shin and Kim, 2016; Xu and Singh, 1998). GR4J operates with four parameters, each representing key hydrological processes. It contains a runoff production store and a routing store. The model calculates effective rainfall by deducting evaporation and interception from precipitation. A portion of this effective rainfall feeds into the production store, while the remainder is partitioned into two flow components routed through triangular unit

hydrographs—one for fast response (UH1) and another for slow response (UH2), each with distinct time bases. Additionally, the model incorporates a groundwater exchange term, which dynamically adds or subtracts water from the routing store to account for subsurface interactions (For details see Perrin et al., 2003).

Unlike the HBV model, the SIMHYD and GR4J models do not include a built-in snowmelt module. However, accurately simulating rainfall-runoff processes in Central Europe requires accounting for snow accumulation and melt. Therefore, an

external degree-day snow module from the Soil and Water Assessment Tool (SWAT, Neitsch et al., 2009) was integrated into both the SIMHYD and GR4J models. Daily snowmelt is then utilized in the rainfall-runoff processes, including



interception, soil moisture, ground water, and runoff generation (For details on the simulation of the rainfall-snowfall separation, snowfall accumulation, and snowmelt see Neitsch et al., 2009 and Li et al., 2014).

## 3.2 Long Short-Term Memory (LSTM) model

Long Short-Term Memory (LSTM) networks are a specialized type of recurrent neural network (RNN), developed to address challenges like gradient explosion and vanishing gradients in traditional RNNs (Hochreiter and Schmidhuber, 1997; Kratzert et al., 2018). LSTMs use gating mechanisms—including input, output, and forget gates, along with a cell state—to regulate information flow, making them particularly effective for time series analysis (Li et al., 2023) and rainfall-runoff modeling (Kratzert et al., 2018).

Given that we focus on simulating extremes, where input data are highly skewed, Min-Max scaling was used for feature preprocessing instead of standardization to better preserve extreme values. Various model configurations were trained and tested via a grid search approach to determine optimal hyperparameters (see appendix A for details). The final network architecture consists of two stacked LSTM layers, each with 128 units, followed by a dense output layer. To reduce overfitting, each LSTM layer employs a dropout rate of 10%. The sequence length was set to 90, with a batch size of 512,

and the training spans up to 120 epochs.

## 3.3 Hybrid models

Our hybrid modeling strategy integrates physics-based hydrological models with a data-driven LSTM model to improve runoff prediction, especially for capturing unprecedented flood events. This approach aims to combine the strengths of both model types to capture the nonlinear patterns in runoff generation by incorporating intermediate hydrological state variables

alongside meteorological inputs. First, each hydrological model is calibrated individually for each catchment. Once calibrated, these models simulate daily runoff, thus generating intermediate state variables that reflect key physical processes that drive floods (see Table A1 for a summary of the intermediate state variables). The LSTM model receives meteorological inputs—consistent with those used as forcing in the hydrological models—and the intermediate state variables from the calibrated hydrological models as input features. By including these state variables, the LSTM leverages physically

meaningful information about catchment conditions, such as snowpack, soil moisture, and groundwater storage, enhancing its capacity to predict complex, nonlinear responses during extreme events. The three hybrid models, each based on a different hydrological model, are hereafter referred to as HBV_LSTM, SIMHYD_LSTM, and GR4J_LSTM, respectively.

## 3.4 Independent flood detection

To evaluate model performance in capturing extreme flood events, both the peak discharge and flood hydrographs of annual

maximum flood events are analyzed. Each event is characterized by a peak value, a start and an end point, following the methodology of Guse et al. (2020). An independent peak is identified if: (1) the lowest discharge between two peaks is smaller than 70% of the smaller peak; (2) the smaller peak is greater than 20% of the annual maximum peak; (3) the



minimum flow between two peaks drops below 20% of the annual maximum flow; (4) the time lag between two peaks is at least 7 days. The event start point is located between the peak and the previous independent peak and the end point is analogously determined (For a detailed description see Guse et al., 2020). Figure 2 shows the annual maximum flood peaks together with their hydrographs and the entire runoff simulation for an exemplary catchment.

## 3.5 Model evaluation and performance metrics

Based on the annual maximum series of observed discharge in each catchment, the 10-year flood peak is estimated using empirical probability analysis and the Weibull formula. Flood events exceeding the 10-year threshold are classified as unprecedented (or out-of-sample) floods, while the remaining events are categorized as in-sample floods. Not all events classified as unprecedented are unprecedented events in the real world, but they are unprecedented for the models, as these events are excluded from the calibration. On average, each catchment experiences three unprecedented flood events based on an average record length of 35 years. This classification enables a comparative assessment of each model's ability to simulate both in-sample and out-of-sample (unprecedented) floods. Figure 2 illustrates the categorized in-sample and out-of-sample flood events in the exemplary catchment.

For model calibration, long-term daily observations are used, excluding the years in which unprecedented floods occurred. After calibration, the models are re-run continuously using the full dataset. Their performance in simulating extreme events is evaluated during the previously excluded periods, referred to as the validation period. To prevent data leakage and ensure that the models are not influenced by prior knowledge from the validation period, a 90-day buffer period is introduced following the validation period. This buffer period is also excluded from model calibration to maintain the integrity of the evaluation.

For the pure hydrological models, the Shuffled Complex Evolution (SCE-UA) algorithm is utilized for parameter optimization (Duan et al., 1993), employing the Nash-Sutcliffe Efficiency (NSE) as the loss function. The LSTM-based models use the Adam optimizer through the Python TensorFlow package (Abadi et al., 2016), with Mean Squared Error (MSE) as the loss function.

The models' performance in simulating runoff and flood events is evaluated using the Nash-Sutcliffe Efficiency (NSE) and Root Mean Square Error (RMSE), which indicate how well the simulated discharge aligns with observed values. Additionally, the relative error between simulated and observed flood peak discharge is used to assess the models' ability to accurately capture flood peaks.

To examine the impact of flood extremity on the models' performance, the discharges are normalized to enable comparison across models and catchments. The observed or simulated discharge ($Q$) is normalized by the maximum observed discharge during the training period ($Q_{\max}$) in each catchment $Q_n = Q/Q_{\max}$. As a result, all normalized in-sample observed flood discharges are smaller than or equal to 1 ($Q_n \leq 1$), while all normalized out-of-sample observed flood peaks exceed 1 ($Q_n > 1$).



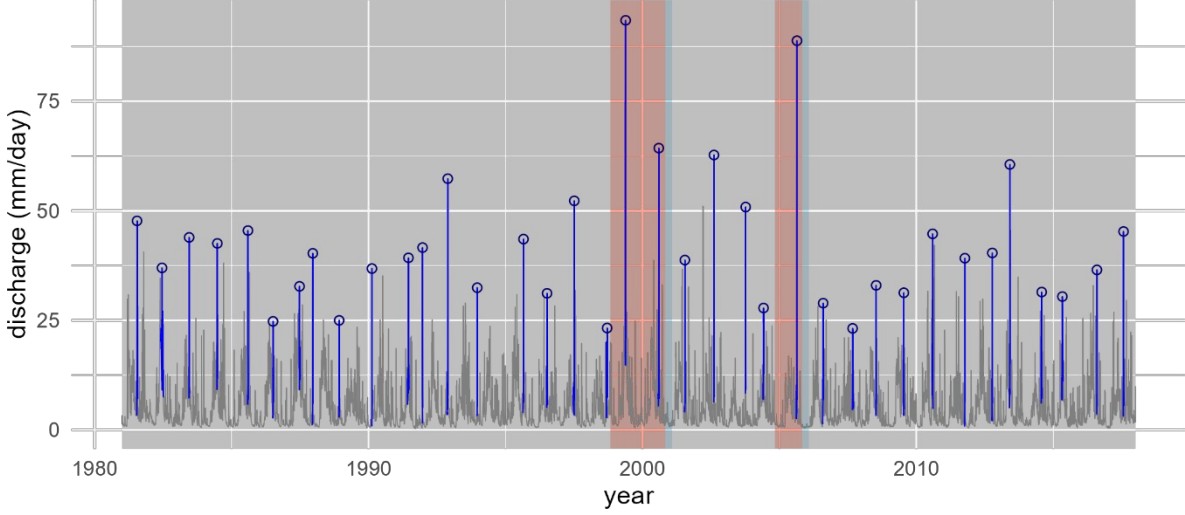

**Figure 2. Observed discharge from 1981 to 2017 for an exemplary catchment. The gray lines represent daily discharge variations, while the blue line segments highlight the hydrographs of the annual maximum flood events, with blue circles marking the annual peak discharges. Periods without a colored background are used for model calibration. The years highlighted in tomato-colored background indicate the validation periods, where peak discharges exceed the 10-year return level threshold. The light-blue background represents a 90-day buffer period following the validation phase to prevent data leakage from the validation set into model calibration.**

## 4 Results and discussions

### 4.1 Model performance for runoff simulation

The three hydrological models, the pure LSTM model, and the three hybrid models are trained and validated using observational data from 416 catchments in the study area (Figure 3 and Table B1). Among the three hydrological models, GR4J demonstrates the best performance in runoff simulation, with a median NSE of 0.60 during calibration and 0.61 during validation, outperforming both the HBV (0.49/0.52) and SIMHYD (0.46/0.44) models. GR4J also has the lowest RMSE values among the three, further confirming its superior runoff simulation capability. The LSTM model outperform the three hydrological models in runoff modeling, achieving higher NSE and lower RMSE values during both the calibration and validation periods. By incorporating the intermediate variables in the rainfall-runoff simulation, the hybrid models achieved further improvement across most catchments in the study areas, surpassing both the pure data-driven LSTM model and the pure hydrological models (Figure B1-6). This enhancement is particularly notable for the hydrological models HBV and SIMHYD that initially exhibited lower performance. Overall, the hybrid models, which incorporate additional hydrological information, effectively enhance the simulation of the rainfall-runoff process.



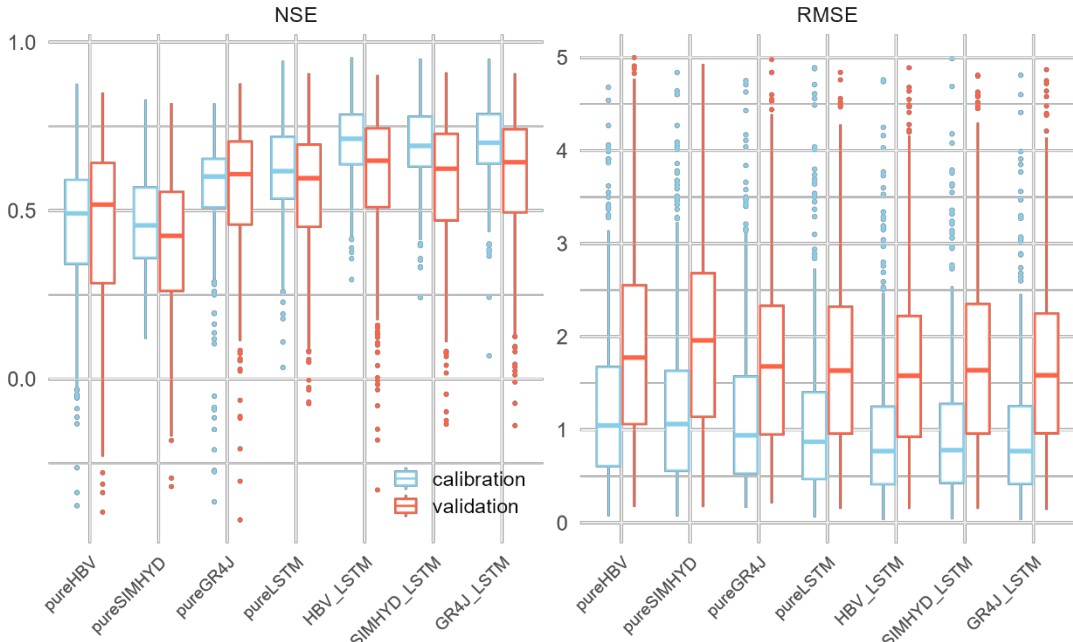

**Figure 3. Comparison of runoff simulation performance across different models in terms of NSE and RMSE (mm/d).**

**4.2 Model performance for unprecedented floods**

The models' performance in reproducing both unprecedented (out-of-sample) and normal (in-sample) floods is evaluated
using NSE, RMSE, and the relative error between simulated and observed flood peaks (Figure 4). Overall, the models' ability
to reproduce flood events aligns with their performance in simulating the general rainfall-runoff behavior. Among the
models, the hybrid models demonstrate the best performance, followed by the pure LSTM model, while the GR4J models
outperform both HBV and SIMHYD models. The three hydrological models show a comparable performance in simulating
both in-sample and out-of-sample flood events, as indicated by the interquartile ranges (IQR) of NSE where in-sample flood
simulations generally overlap with out-of-sample floods. However, RMSE values for out-of-sample floods were consistently
higher across all models, likely due to the higher discharge magnitudes in these events, which lead to larger absolute
simulation errors despite similar NSE values. The hybrid models generally outperform their corresponding hydrological
models in capturing in-sample floods, but their advantage diminish when simulating out-of-sample floods. Notably, the
GR4J model exhibits competitive performance with the data-driven models in out-of-sample flood simulations, achieving
similar median NSE and RMSE values (Table B 2).

In terms of flood peak simulation, all models tend to underestimate flood peaks by over 30% for most in-sample events and
by over 50% for most unprecedented (out-of-sample) events (Figure 4, Table B 2). This result shows a substantial
performance decline when the models are used for extrapolation to rare events. For in-sample floods, the hybrid models
outperform the corresponding pure hydrological models, showing less underestimation. However, for out-of-sample floods,





the differences in peak reproduction across models are minor, with both the hydrological models and the hybrid models showing similar performance. The underestimation of in-sample flood peaks may be partly due to issues with the quality of rainfall and discharge data or catchment-specific characteristics not fully captured by the models. The challenges in simulating out-of-sample floods are likely a result of these events not being included in model calibration or training, as well as the inherent limitations in the model structures, which hinder their ability to fully capture the patterns of these extreme

events.

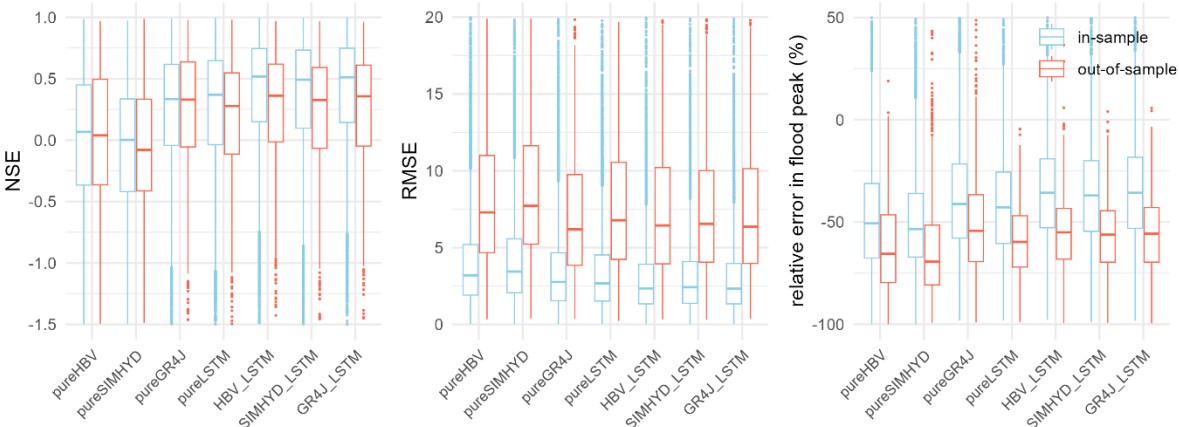

**Figure 4. Comparison of simulation performance for unprecedented (out-of-sample) and in-sample floods across different models in terms of NSE, RMSE (mm/d) and relative error in flood peaks.**

**4.3 Effect of flood extremity on model performance**

To evaluate how flood extremity affects model performance in simulating floods, Figure 5 compares normalized flood peak discharges with the model performance metrics, specifically the difference in absolute relative error between simulated and observed flood peaks ($\Delta|Re\_peak|$. Additionally, Figure 6 compares simulated and observed normalized flood peaks across different models.

For in-sample floods (with normalized peaks not greater than 1), the hybrid models consistently outperform the pure LSTM

model. However, no significant relationship is observed between the performance differences of these models and flood magnitude. The local trends estimated using the LOESS method (Figure 5) indicate no clear correlation pattern, which is also reflected in Figure 6. Compared to the pure hydrological models, the hybrid models perform better in simulating floods of medium magnitude (normalized peaks between 0.5 and 1.0), particularly for the HBV and SIMHYD models. Both of these hydrological models initially show weaker performance in simulating the rainfall-runoff process compared to the GR4J

model.

For unprecedented (out-of-sample) floods with normalized discharge peaks greater than 1, the hybrid models generally outperform the pure hydrological models in capturing flood peaks, particularly for less extreme events (normalized peaks between 1 and 1.5). However, the benefits of hybridization diminish rapidly when normalized peaks exceeded 2. In these





cases, the hybrid models consistently underestimate observed flood peaks (Figure 6). Additionally, the simulated normalized
flood peaks for out-of-sample extremes are mostly not greater than 1 (the historical maxima in the training period),
suggesting a potential extrapolation limit in LSTM-based models (Figure 6).

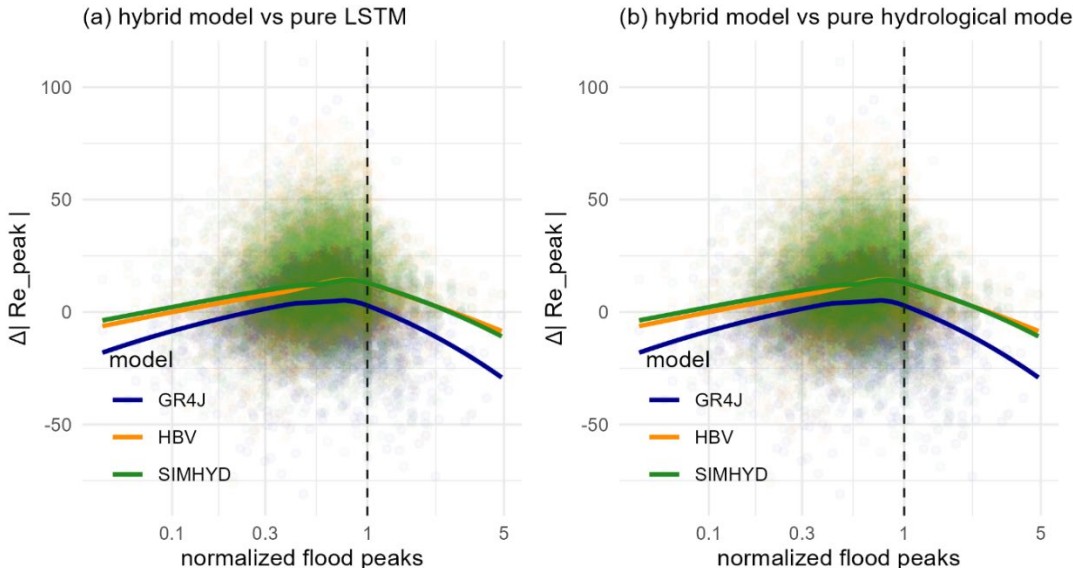

**Figure 5 Relationship between normalized flood peaks and difference in absolute relative error in flood peak simulation
($\Delta|Re\_peak|$) across hybrid, pure LSTM and pure hydrological models for the unprecedented (out-of-sample) floods. Positive
values of $\Delta|Re\_peak|$ indicate that the hybid models showed improved simulation performance compared to the pure LSTM
model or the correspoding hydrological models. The solid lines represent local trends estimated using the LOESS (Locally
Estimated Scatterplot Smoothing) method, a smoothing technique that fits local regressions to capture patterns in the data, for
each of the three hydrological models.**





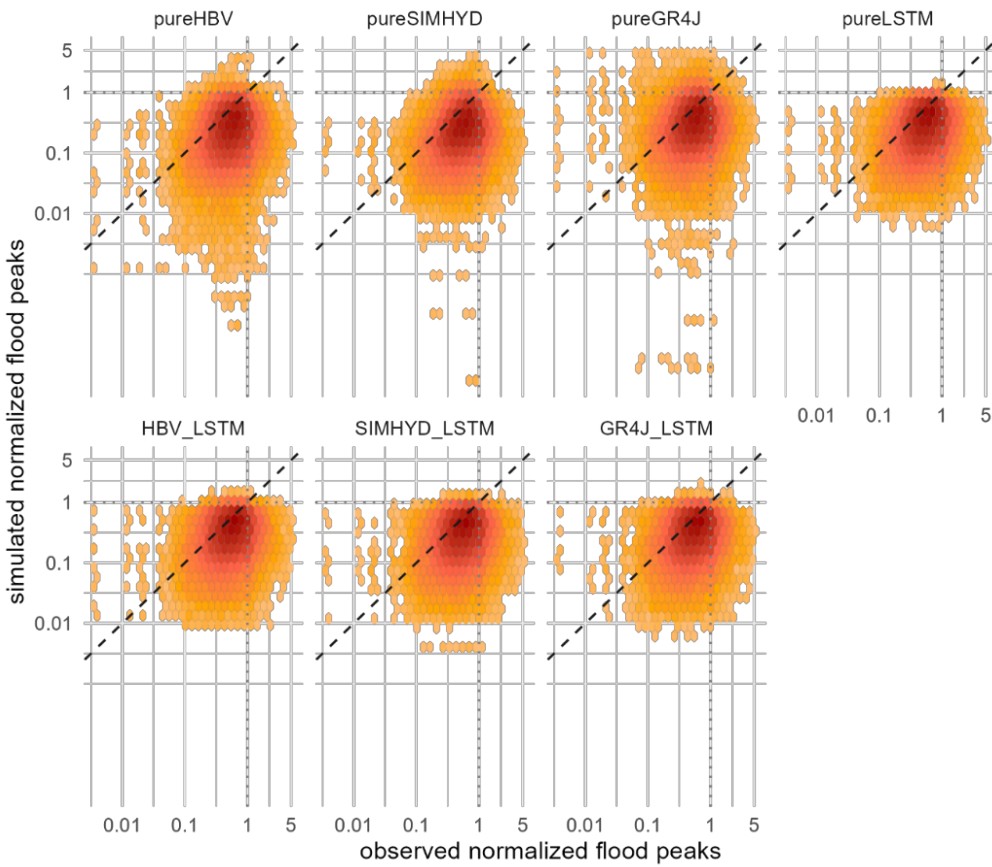

**Figure 6. Comparison between simulated and observed normalized flood peaks across different models. The dashed line represents a 1:1 slope. The point colors indicate data density, with red representing high density.**

## 4.4 Discussion

The results of this study suggest that hybrid models offer notable improvements in simulating discharge and in-sample flood events in Central Europe compared to pure hydrological models and the purely data-driven LSTM model. The extent of these improvements varies with the structure of the underlying hydrological model. Specifically, hybrid models based on HBV and SIMHYD showed greater enhancement than the GR4J-based hybrid model. This observation raises critical questions about the mechanisms driving the improvements in hybrid models and the factors that influence this enhancement. One possible explanation is that compared to the GR4J model, HBV and SIMHYD may leave more "room" for the data-driven component of the hybrid model to adjust and compensate for their structural limitations. However, when paired with LSTM, which can learn dependence patterns directly from data, these models might benefit more, as LSTM can augment or correct aspects of the flood dynamics that are otherwise missed.

The extrapolation limitations of LSTM-based models, particularly for unprecedented floods exceeding the bounds of training data, may stem from inherent structural and statistical constraints. The bounded activation functions (e.g., tanh and sigmoid,



see Figure A1) saturate during extreme events, truncating the model's sensitivity to inputs beyond historical maxima, while

normalization techniques like MinMax scaling used in this study implicitly tie predictions to the training data's range (Kratzert et al., 2024). These limitations are compounded by the absence of physical principles governing hydrological systems, such as mass balance or energy conservation. We suggest synthetic design storms (Macdonald et al., 2025) can be generated carefully to demonstrate how hidden states in LSTMs saturate near the activation function bounds, capping the model's capacity to represent unprecedented events. Specifically, we can simulate a series of increasing storm intensities

beyond the training range, evaluate the model's response to these inputs by tracking activation values in hidden states, and examine whether LSTM predictions plateau due to activation function saturation. This approach can provide direct evidence of LSTM extrapolation limits in flood prediction. Additionally, augmenting training data with synthetic extremes, generated via stochastic rainfall-runoff simulations, could expand the model's exposure to tail events. To overcome the lack of physical constraints, future work should prioritize tighter coupling of data-driven and physics-based modeling approaches.

Differentiable hydrological models (Feng et al., 2023; Kapoor et al., 2023) could enforce physical laws during training, while physics-guided loss functions might penalize implausible states, such as a negative water storage. Our current hybridization strategy, which integrates hydrological states, lays a foundation for such advancements, but deeper integration of physical equations and causal drivers—rather than state variables concatenation—could further bridge the gap between empirical flexibility and hydrological plausibility. Ultimately, overcoming extrapolation limits will require models that not only learn

from data but also reason through physics, even when confronting the unknown.

In this study, models were implemented independently for each test catchment, without considering the potential benefits of a regional approach. A regional modeling strategy, such as "entity-aware" LSTM models, could integrate static catchment attributes that capture similarities and differences in runoff-generation mechanisms across catchments (Kratzert et al., 2019a; Kratzert et al., 2019b). Pooling information from multiple catchments may allow flood data from one catchment to inform

simulations in others, including for floods that might be unprecedented in a given catchment but historically observed in another. This "augmentation effect" could enhance the predictive accuracy of hybrid models by leveraging spatially diverse yet hydrologically similar flood data. Such a regional approach could be especially valuable in improving the hybrid models' ability to simulate unprecedented events by using data from a broader range of historical conditions within the region. However, this approach raises an additional challenge: if an unprecedented flood were to occur simultaneously across the

entire region, the model would still lack prior data to draw from, similar to the issue faced when simulating unprecedented events at the local scale. This limitation underscores the importance of examining how well regional approaches can generalize to truly novel flood events that occur across multiple catchments, as well as the role that diverse but regionally relevant data plays in improving predictions.



## 5 Conclusions

In this study, we evaluated the performance of three hydrological models, a deep-learning Long Short-Term Memory (LSTM) network, and three hybrid models in simulating unprecedented floods. The hybrid models combine rainfall-runoff process variables from hydrological simulations with meteorological inputs to improve runoff simulation. Here, an unprecedented flood is defined as the event with peak discharge exceeding the 10-year flood, referred to as out-of-sample since it was excluded from the model training. Annual maximum floods with peaks below the 10-year flood were
categorized as in-sample for model performance comparison. The pure deep-learning LSTM model generally outperformed traditional hydrological models across most catchments in the study area, demonstrating strong capability in runoff simulation. The hybrid models further improved rainfall-runoff modeling performance, leveraging the strengths of both hydrological and data-driven components. However, when modeling flood events, all models—including pure hydrological, pure LSTM, and hybrid—tended to underestimate peak discharges for both out-of-sample (unprecedented) and in-sample
floods. Notably, the hybrid models showed reduced underestimation. Yet, as flood magnitude increased to more extreme levels, the performance advantage of the hybrid models diminished rapidly. An extrapolation bound was observed in LSTM based models which is probably attributed to inherent structural and statistical constraints. Our analysis suggests that LSTMs struggle to generalize when confronted with flood extremes surpassing historical maxima, underscoring the need for methodological advancements.

To improve extrapolation, future work should focus on tighter integration of physical principles and deep learning models. This includes differentiable hydrological models to enforce mass and energy conservation, physics-guided loss functions to prevent implausible states, and synthetic extreme event generation to expand training data exposure. Additionally, regional modeling approaches could leverage hydrological similarities across catchments to improve predictions of extreme events, even in locations with limited historical data. Ultimately, combining data-driven learning with hydrological reasoning will be
key to developing models that remain reliable even under extreme, unseen conditions.

## Appendix A Long Short-Term Memory (LSTM) model

### A1 LSTM model architecture

An LSTM network is a type of recurrent neural network (RNN) with dedicated memory cells that store information over long periods. It uses gated mechanisms to control information flow, making it well-suited for modeling dynamical systems
like watersheds. Unlike standard RNNs, LSTMs mitigate exploding and vanishing gradient issues, enabling them to learn long-term dependencies effectively.

An LSTM works as follows: given an input sequence $x = \left[ x[1], \cdots, x[T] \right]$ with $T$ time steps, where each element $x[T]$ is a vector containing input features (model inputs) at time step $t$ ($1 \leq t \leq T$), the following equations describe the forward pass through the LSTM:






$$i[t] = \sigma(\mathbf{W}_i x[t] + \mathbf{U}_i h[t-1] + b_i)$$
$$f[t] = \sigma(\mathbf{W}_f x[t] + \mathbf{U}_f h[t-1] + b_f)$$
$$g[t] = \tanh(\mathbf{W}_g x[t] + \mathbf{U}_g h[t-1] + b_g)$$
$$o[t] = \sigma(\mathbf{W}_o x[t] + \mathbf{U}_o h[t-1] + b_o)$$
$$c[t] = f[t] \odot c[t-1] + i[t] \odot g[t]$$


$$h[t] = o[t] \odot \tanh(c[t])$$

where $i[t]$, $f[t]$, and $o[t]$ are the input gate, forget gate, and output gate, respectively, $g[t]$ is the cell input and $x[t]$ is the network input at time step $t$ ($1 \leq t \leq T$), and $h[t-1]$ is the recurrent input, $c[t-1]$ the cell state from the previous time step. After the first time step, the hidden can cell states are initialized as a vector of zeros. $\mathbf{W}$, $\mathbf{U}$, and $b$ are learnable parameters for each gate, where subscripts indicate which gate the particular weight matrix/vector is used for, $\sigma(\cdot)$ is the

sigmoid function, $\tanh(\cdot)$ is the hyperbolic tangent function, and $\odot$ is element-wise multiplication. Function curves for $\sigma(\cdot)$ and $\tanh(\cdot)$ can be seen in Figure A1.

The intuition behind this network is that the cell states ($c[t]$) characterize the memory of the system. The cell states can get modified by the forget gate ($f[t]$), which can delete states, and the input gate ($i[t]$) and cell update ($g[t]$), which can add new information. In the latter case, the cell update is seen as the information that is added and the input gate controls into

which cells new information is added. Finally, the output gate ($o[t]$) controls which information, stored in the cell states, is outputted.

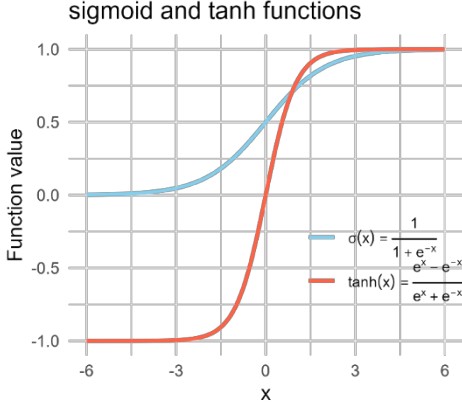

**Figure A 1. Function curves of sigmoid $\sigma(\cdot)$ and $tanh(\cdot)$ in LSTM model.**

**A2 Hyperparameter determination in LSTM model**

The hyperparameters in the setup of LSTM, i.e., the number of hidden states (unit length), dropout rate, length of the input sequence, and number of stacked LSTM layers, were derived by running a grid search over a range of parameter values. Concretely we considered the following possible parameter values.

unit length: 24, 64, 96, 128, 156, 196



dropout rate: 0.1, 0.2, 0.4, 0.5

length of input sequence: 90, 180, 270, 365

number of stacked LSTM layer: 1, 2

The dataset from each individual catchment within the research domain were spilt and then combined together into training

and validation sets. Each hyperparameter combination was trained and validated. The distributions of NSE values from

simulations using various hyperparameter combinations are shown in Figure A 2. The final configuration was chosen by

taking the parameter set that resulted in the highest median NSE over all possible parameter configurations. The derived

parameters are the following.

unit length: 128

dropout rate: 0.1

length of input sequence: 180

number of stacked LSTM layer: 2

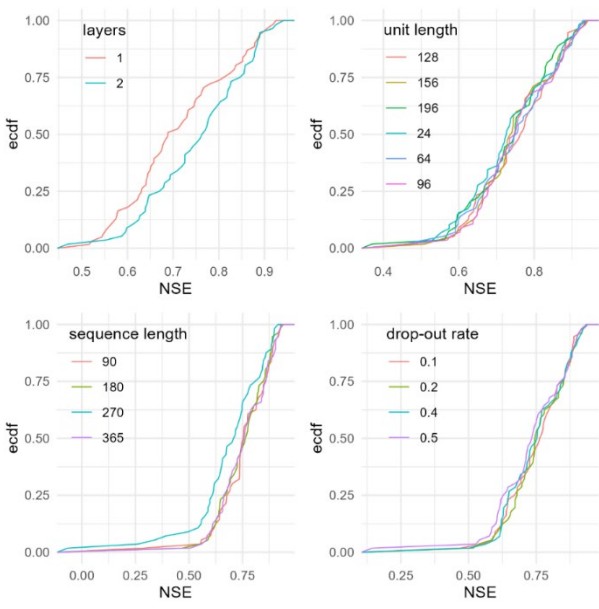

**Figure A 2. Comparison of empirical cumulative distribution functions (ecdf) of NSE values in runoff simulation using the LSTM models with different hyperparameter combinations. In each subplot, only the specified hyperparameter was varied, while all other parameters remained fixed at their optimal values.**






## A3 Hybrid model construction

**Table A 1. Overview of the intermediate variables simulated from 3 hydrological models integrated to hybrid modeling**

| Hydrological model | Intermediate variable | Explanation |
|---|---|---|
| HBV | Snow | Water equivalent of snowpack |
| | Snow_melt | Snow melt |
| | S_SOIL | Water content in soil box |
| | S_UZ | Water content in upper groundwater box |
| | S_LZ | Water content in lower groundwater box |
| SIMHYD | Snow | Water equivalent of snowpack |
| | Snow_melt | Snow melt |
| | SMS | Soil water content (moisture) |
| | GW | Underground water content |
| GR4J | Snow | Water equivalent of snowpack |
| | Snow_melt | Snow melt |
| | Pr | Total quantity of water reaching routing functions |
| | S | Water content in production storage |
| | R | Water content in routing storage |

## Appendix B

**Table B 1. Comparison of model performance in streamflow simulation during calibration and validation periods across the tested 416 catchments in central Europe with two metrics: NSE and RMSE (mm/d). Median and mean columns indicate the median and mean values of**

| model | period | NSE | | RMSE (mm/d) | |
|---|---|---|---|---|---|
| | | median | mean | median | mean |
| pureHBV | calibration | 0.490 | 0.439 | 1.04 | 1.34 |
| | validation | 0.519 | 0.452 | 1.86 | 2.11 |
| pureSIMHYD | calibration | 0.458 | 0.466 | 1.07 | 1.35 |
| | validation | 0.442 | 0.409 | 2.02 | 2.24 |
| pureGR4J | calibration | 0.601 | 0.549 | 0.95 | 1.24 |
| | validation | 0.608 | 0.559 | 1.75 | 1.92 |
| pureLSTM | calibration | 0.616 | 0.622 | 0.875 | 1.14 |
| | validation | 0.594 | 0.562 | 1.67 | 1.94 |
| HBV_LSTM | calibration | 0.714 | 0.709 | 0.77 | 0.995 |
| | validation | 0.649 | 0.600 | 1.62 | 1.86 |
| SIMHYD_LSTM | calibration | 0.692 | 0.694 | 0.785 | 1.02 |
| | validation | 0.624 | 0.583 | 1.68 | 1.9 |
| GR4J_LSTM | calibration | 0.702 | 0.707 | 0.77 | 0.998 |
| | validation | 0.643 | 0.601 | 1.63 | 1.86 |





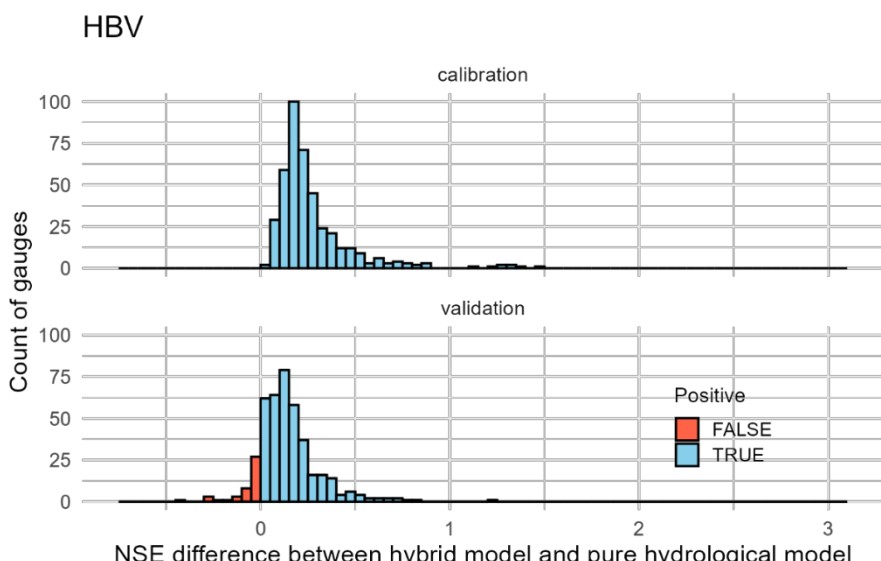

**Figure B 1. Distribution of NSE differences between the hybrid model (HBV-LSTM) and the pure hydrological (HBV) model in stream simulation across test catchments.**

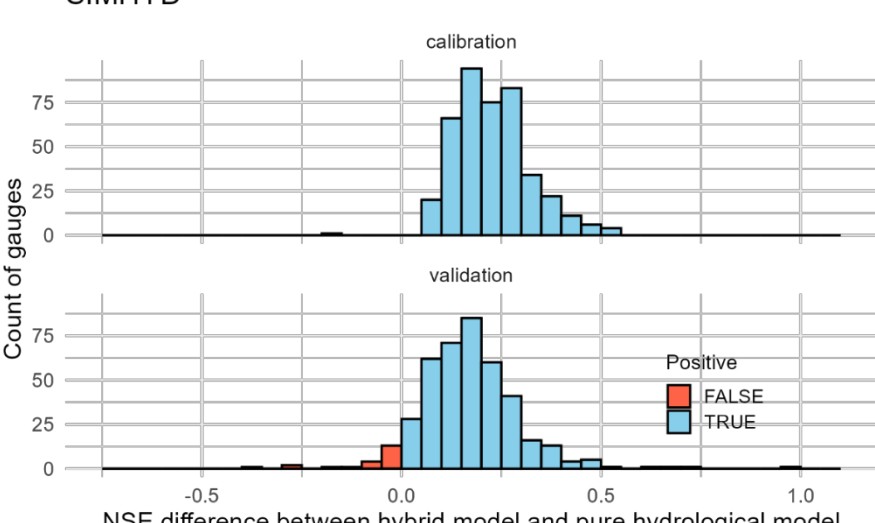

**Figure B 2. Distribution of NSE differences between the hybrid model (SIMHYD-LSTM) and the pure hydrological (SIMHYD) model in stream simulation across test catchments.**



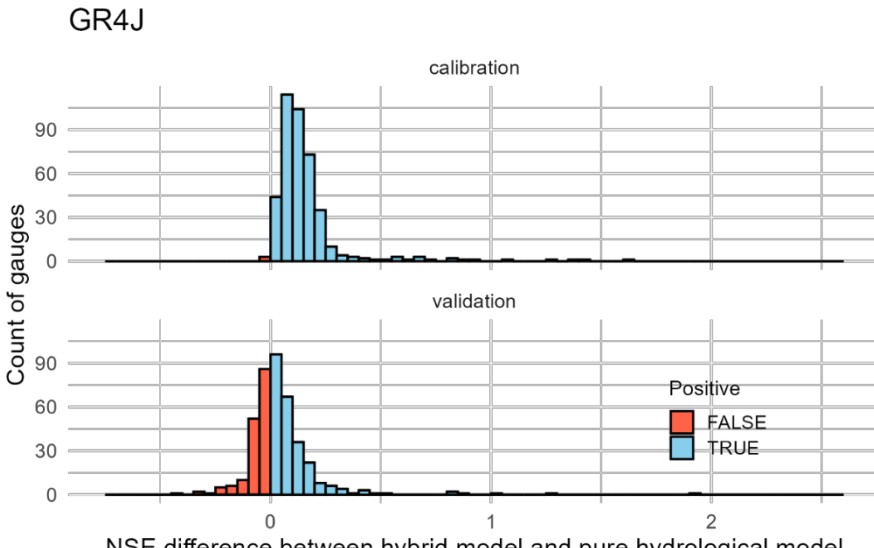

**Figure B 3. Distribution of NSE differences between the hybrid model (GR4J-LSTM) and the pure hydrological (GR4J) model in stream simulation across test catchments.**

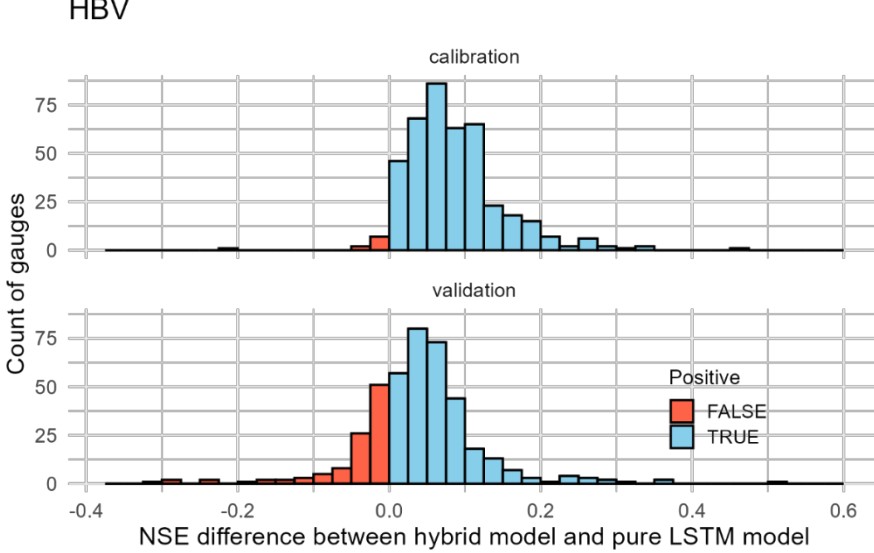

**Figure B 4. Distribution of NSE differences between the hybrid model (HBV-LSTM) and the pure LSTM model in stream simulation across test catchments.**




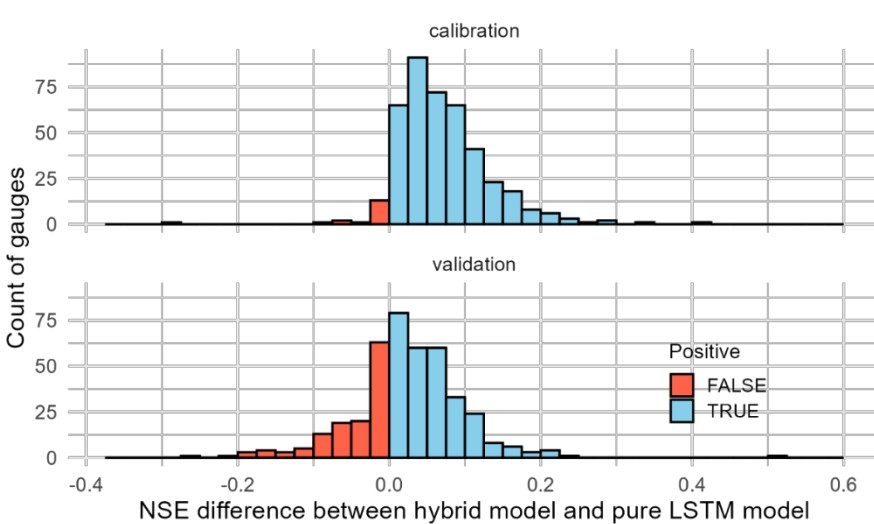

**Figure B 5. Distribution of NSE differences between the hybrid model (SIMHYD-LSTM) and the pure LSTM model in stream simulation across test catchments.**

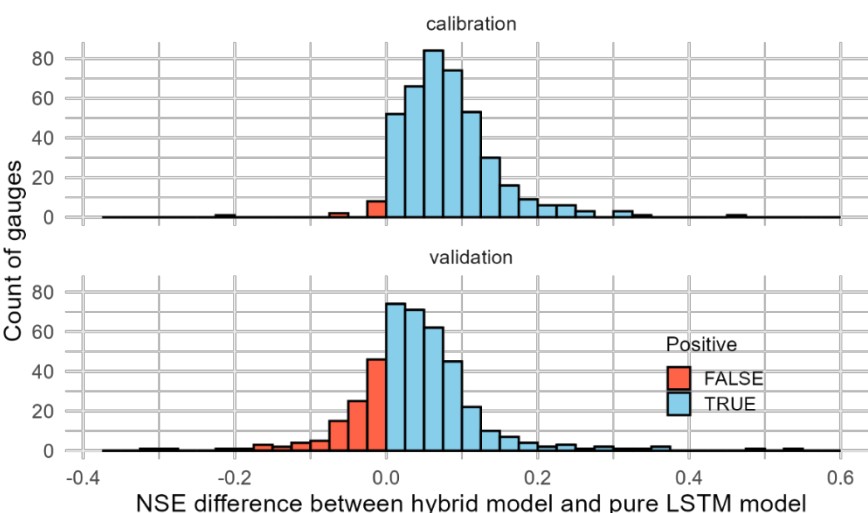

**Figure B 6. Distribution of NSE differences between the hybrid model (GR4J-LSTM) and the pure LSTM model in stream simulation across test catchments.**





**Table B 2. Performance comparison of models in simulating flood discharges in terms of 3 metrics: NSE, RMSE (mm/d) and relative error between simulated and observed flood peaks (Re_peak, %). Median and mean columns indicate the median and mean values of the metrics among all the catchments.**

| model | flood | NSE | | RMSE (mm/d) | | Re_peak (%) | |
|---|---|---|---|---|---|---|---|
| | | median | mean | median | mean | median | mean |
| pureHBV | in-sample | 0.051 | -0.029 | 3.22 | 4.28 | -50.6 | -47.6 |
| | out-of-sample | 0.028 | -0.016 | 7.75 | 9.74 | -65.6 | -62.7 |
| pureSIMHYD | in-sample | -0.015 | -0.107 | 3.45 | 4.40 | -53.3 | -49.4 |
| | out-of-sample | -0.099 | -0.123 | 8.16 | 10.02 | -69.2 | -62.6 |
| pureGR4J | in-sample | 0.330 | 0.222 | 2.78 | 3.74 | -41.0 | -38.1 |
| | out-of-sample | 0.328 | 0.231 | 6.55 | 8.40 | -54.3 | -51.2 |
| pureLSTM | in-sample | 0.366 | 0.249 | 2.68 | 3.62 | -42.8 | -41.9 |
| | out-of-sample | 0.275 | 0.175 | 7.17 | 8.88 | -59.7 | -59.3 |
| HBV_LSTM | in-sample | 0.517 | 0.388 | 2.34 | 3.16 | -35.6 | -35.5 |
| | out-of-sample | 0.361 | 0.251 | 6.71 | 8.39 | -55.0 | -55.4 |
| SIMHYD_LSTM | in-sample | 0.490 | 0.357 | 2.42 | 3.24 | -36.9 | -36.8 |
| | out-of-sample | 0.323 | 0.221 | 6.84 | 8.58 | -56.2 | -56.9 |
| GR4J_LSTM | in-sample | 0.511 | 0.385 | 2.34 | 3.17 | -35.6 | -35.3 |
| | out-of-sample | 0.357 | 0.237 | 6.68 | 8.45 | -55.7 | -55.9 |

**Author contribution**

XG, BS and BM developed the concept; XG and BM designed the research experiments; XG developed and validated the models; XG wrote the paper with contributions from all the authors.

**Data availability**

The used hydrometeorological datasets for the test catchments in our study can be sourced on Zenodo, LamaH-CE: LArge-SaMple DAta for Hydrology and Environmental Sciences for Central Europe https://zenodo.org/records/5153305 (last access: 04 November 2024).

**Code availability**

The hydrograph extraction algorithm, used to identify independent flood events from daily discharge observations, is available in the GitHub repository "Flood_hydrograph"(https://github.com/XiaoxiangGuanGFZ/Flood_hydrograph.git).



The codes used to implement and validate the models (GR4J, HBV, SIMHYD and LSTM) in rainfall-runoff simulation are available on Zenodo (https://doi.org/10.5281/zenodo.14049563, Guan, 2024)

**Competing Interest**

The authors declare that they have no known competing financial interests or personal relationships that could have appeared to influence the work reported in this paper.

**Acknowledgement**

Xiaoxiang Guan is funded by the China Scholarship Council for his PhD research (Grant #: 202106710029).

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
