# Peer review of "Beyond Observed Extremes: Can Hybrid Deep Learning Models Improve Flood Prediction?"

_EGUsphere, 2025_

## Referee Comment (RC1)

**Review of HESS Manuscript**

*"Beyond Observed Extremes: Can Hybrid Deep Learning Models Improve Flood Prediction?"*

Please find attached my review of the manuscript.

**1. Scope**

The scope of the article is inside the scope of HESS.

**2. Summary**

The authors make a comparison study between conceptual, data-driven and hybrid models to evaluate the generalization capabilities of the different models. They test their methods in a subset of the Lahma CE dataset.

**3. General evaluation**

Even though the authors present interesting ideas, there are important points that need to be considered.

The literature review needs to be improved.

- The hybrid model architecture they are using was proposed by Frame2021, and this was not mentioned.
- The experimental setup you are using (training/test split based on probability of the data) was proposed by Frame2022, and this was not mentioned.
- Acuna2025 did extremely similar experiments to evaluate the generalization capabilities of hybrid models, and this was also not mentioned.

Using the same strategy and experimental setup as previous studies and applying it on a new dataset is a valid approach and can help generalize results. However, it should be mentioned as such. Moreover, the results should be further compared with exiting studies to better place the study with current literature (more details are given below).

A second point that needs to be further clarified is if the data-driven methods (both lstms and hybrid) were trained locally for each basin or regionally for all basins at once, because this could have major implications in the results that are being presented. According to line 286 the models were trained independently for each catchment (I apologize if I misunderstood). If this was also the case for the lstm and the hybrid models, the result would be biased, as extensive literature has shown that data-driven methods should be trained regionally.

Lastly, I believe the hybrid model architecture chosen by the author might not be ideal to answer their research questions. By having the LSTM after the conceptual model, the hybrid model will experience the same problems as the LSTM, and the mass-conservative structure of the conceptual part is not being utilized to overcome the extrapolation issues that the data-driven part might present. Other issues with respect to model intercomparison are detailed below

**4. Specific comments**

Line 47: This part should be further explained. LSTMs work much better when trained regionally (Kratzert2024). Thus, they can generalize from one basin to another and predict discharges for a specific basin beyond the levels observed during training (for that basin). They do struggle in generalizing above the maximum overall discharge saw during training, and this has been studied by Baste2025.

What I also think is missing here is a reference to Frame2022, which did experiments to evaluate the generalization capabilities of LSTM and process-based models, using the same methodology that you are using (train in high-probability years and test on low-probability ones). This study is cited below in line 58, but not in the correct context, because he did not use hybrid models, and he did use metrics to evaluate specifically high flow scenarios. Given that you are using their methodology, further discussion on Frame2022 is needed in the introduction.

Line 59-64: Here you are stating as a literature gap that the extrapolation capabilities of hybrid models have not been explored, but this is not true. Acuna2025 did an extremely similar study where he evaluated the generalization capabilities of hybrid models, compared to LSTMs and process-based models, by training on high-probability years and testing on low-probability ones. Also, Baste2025 evaluated the extrapolation capabilities on LSTM and hybrid models by evaluating their performance on synthetic events.

Some differences of this study with the previous studies is the dataset in which you are running your experiments and the hybrid architecture that is being used. But you should be more specific in the literature gap you are filling.

Line 90:  AcuñaEspinoza2024 did not used the HBV model.

Line 91: What do you mean by "Using daily or hourly time steps, the model is forced by precipitation, air temperature, and monthly estimates of potential evapotranspiration"?  I agree that the model can be run in daily resolution or in hourly resolution, but the PET should also be consistent with the temporal scale of the other variables. Why are you referring to monthly estimated of PET?

Section 3.1: It would be nice if you add a figure in appendix with the different structures of the conceptual models.

Line 120: You can reformulate to "exploding and vanishing gradients..."

Line 125: Why would the Min-Max scaling better preserve extreme values? Also, did you calculate the scaler based only on the training period?   Was the LSTM trained regionally?

Section 3.3:

Using LSTMs as a postprocessor of process-based models was introduced by Frame2021. You should indicate this.

In this section you indicate that the process-based models are trained individually. Is the LSTM postprocessor trained regionally or are you training one per basin? Does it receive static attributes?

Are you training the hybrid model seq-to-seq or seq-to-1? And if it is the first case, what sequence length are you using? These details are quite important.

Furthermore, there is something that you should consider with the hybrid model architecture you are using. Kratzert2024, Acuna2025 and Baste2025 have talked about the theoretical saturation limit that LSTMs have, and the former two discussed their effect in extrapolation. Moreover, both of these studies evaluate hybrid models to see if they can partially mitigate the saturation limit. However, for this to work the conceptual model must be after the LSTM. If the process-based model is before the LSTM, the hybrid model will still suffer from the LSTM saturation, and the process-based part cannot really contribute to extrapolate. Therefore, the hybrid model architecture you are using is not ideal if what you are interested in is extrapolation capabilities.

Moreover, in line 69 of the introduction you indicate that you are interested in exploring the effects of model structure. Having the LSTM after the process-based model will not allow you to do this, as the LSTM is flexible enough to modify the signals coming from the process-based models and correct for structural deficiencies in the model structure. Acuna2024 tested this for hybrid models where the LSTM is before the process-based model, and I would argue that if the LSTM is after the conceptual models, the compensation effect can be even larger.

In summary, even though you are presenting a nice set of experiments, I believe that the hybrid model architecture you chose will not allow you to properly answer your research questions.

Line 161: What do you mean by long-term daily observations? Are the stand-alone LSTMs trained seq-seq?

Line 164: The idea of the buffer period was proposed by Frame2022. You should cite this here.

Line 169: The basin-averaged NSE proposed by Kratzert2019b has shown good results because it avoids overweighting humid basins, and it has been the standard in multiple benchmarks (Kratzert2019b, Lees2021, Loritz2024). Did you evaluate if this loss function gives better results for your case?

Line 198: Did you benchmark your LSTM implementation in other datasets (CAMELS-US or CAMELS-GB)? This is especially important if you are using your own implementation of the models, to assure that your model is functioning correctly. I ask this because the median value you are reporting for the LSTM (around 0.6) seem a bit low. Gauch2024 reported a median NSE close to 0.72 for a LSTM model evaluated in the Lamah-CE dataset (see figure below). I understand that the training and validation periods are different, Gauch was doing a temporal split, while you are doing a low flows/high flows split. However, Acuna2025 did the low flow/ high flow slip in CAMELS-US and the median NSE they reported (0.75) is quite close to the original NSE (0.76) reported by Kratzert2019b using a temporal training-testing split.

I would suggest that you benchmark your model against existing results found in literature, as this will allow you to show that your LSTM implementation is working up to standards, and further validate your results. You can put the benchmark results in an appendix, as it is not the main point of the paper.

**Experiments**

[Figure]

Source: Gauch, 2024.

Line 205: You could remove the NSE metric when you evaluate the peaks. The RMSE and the Relative Error do make a lot of sense, but the NSE is not telling you much, because the mean discharge used in the denominator of the NSE metric is not a good null hypothesis if what you are interested in is the peaks.

Line 220-225: All of what you are describing here has been previously reported in other studies. It is important to link your results with existing literature to see if they are consistent or not, and if they are not, delve into why this is the case.

Section 4.3

I apologize, but I had a lot of trouble understanding Figure 5. I believe the figure is trying to show multiple things at the same time, which makes it quite confusing. If you want to compare how the different models perform for increasingly higher events, maybe you can use a boxplot that compare the models for different categories of floods (similar to the one presented by Frame2022 and Acuna2025). Moreover, there is a small typo on how you wrote hybrid in Figure´s 5 captions.

Line 244-246: This result should be further discussed, because, if trained regionally, the LSTM should be capable of producing for a specific basin values higher to what it saw during training for that specific basin. This might be a problem related to how you standardized your data (line 175-179). If you divide the discharges by the maximum value saw during training for each basin, the LSTM never sees a target variable larger than 1, which I believe is not a good strategy. A better strategy would be to transform the discharge to mm/h using the catchment area. This strategy has been applied in the benchmark studies for CAMELS US (Kratzert2019b), CAMELS GB (lees2021), CAMELS DE (Loritz2024), and also in other extrapolation experiments (Acuna2025, Baste2025, Frame2022). By mapping everything to mm/h you reduce the relative differences between a small and a large basin, but you are not hard restricting that all the values should be smaller than 1.

Line 263-265: It has been shown before the LSTM can compensate for structural deficiencies on process-based models, so this is not new. However, there are two points that need to be analyzed further. Is the LSTM using the information that the conceptual models are sending him? Frame2021 study this using integrated gradients, and conclude that the LSTM postprocessor was giving higher priority to the meteorological inputs than to what the conceptual model was sending. In their words

"Precipitation inputs were weighted higher than the NWM streamflow output itself, which means that even when NWM streamflow data were available, the LSTM_PPA generally learned to get information directly from forcings rather than from the NWM streamflow output. This indicates that the LSTM_PPA generated a new rainfall–runoff relationship rather than relying on the NWM, which is consistent with the overall results (Figure 2) that showed similar performance between the LSTM_A and LSTM_PPA."

It should be analyzed if this is also the case during high events, because otherwise the conceptual part of the hybrid model is not doing much.

Line 268: It can be also due to the normalization strategy you are using, you should check this.

Line 271-277: This was already done by Baste2025.

Line 282: See comments above about the problems your hybridization strategy might have for extrapolation to extreme events.

Line 286-287: Did you also train the LSTM and the hybrids independently for each basin? Please clarify this, because this would be a huge limitation for the study and can biased the results, specially if what you are interested in is generalization capabilities. The hypothesis that you indicate in line 289-291:

"Pooling information from multiple catchments may allow flood data from one catchment to inform simulations in others, including for floods that might be unprecedented in a given catchment but historically observed in another."

is totally correct, and it has been shown before. Data-driven methods should be trained regionally. Please clarify this.

References:

Acuña Espinoza, E., Loritz, R., Álvarez Chaves, M., Bäuerle, N., & Ehret, U. (2024). To bucket or not to bucket? Analyzing the performance and interpretability of hybrid hydrological models with dynamic parameterization. Hydrology and Earth System Sciences, 28(12), 2705–2719. https://doi.org/10.5194/hess-28-2705-2024

Acuña Espinoza, E., Loritz, R., Kratzert, F., Klotz, D., Gauch, M., Álvarez Chaves, M., & Ehret, U. (2025). Analyzing the generalization capabilities of a hybrid hydrological model for extrapolation to extreme events. Hydrology and Earth System Sciences, 29(5), 1277–1294. https://doi.org/10.5194/hess-29-1277-2025

Baste, S., Klotz, D., Espinoza, E. A., Bardossy, A., and Loritz, R.: Unveiling the Limits of Deep Learning Models in Hydrological Extrapolation Tasks, EGUsphere [preprint], https://doi.org/10.5194/egusphere-2025-425, 2025.

Frame, J.M., F. Kratzert, A. Raney II, M. Rahman, F.R. Salas, and G.S. Nearing. 2021. " Post-Post-Processing the National Water Model with Long Short-Term Memory Networks for Streamflow Predictions and Model Diagnostics." Journal of the American Water Resources Association 57(6): 885–905. https://doi.org/10.1111/1752-1688.12964.

Frame, J.M., Kratzert, F., Klotz, D., Gauch, M., Shalev, G., Gilon, O., Qualls, L.M., Gupta, H.V. and Nearing, G.S. 2022. Deep learning rainfall–runoff predictions of extreme events. Hydrol. Earth Syst. Sci., 26(13), 3377-3392. doi:470 https://doi.org/10.5194/hess-26-3377-2022.

Gauch, M., Kratzert, F., Dube, V., Gilon, O., Klotz, D., Metzger, A., Nearing, G., Ofori, F., Shalev, G., Shenzis, S., Tekalign, T., Weitzner, D., Zlydenko, O., and Cohen, D.: Deep Learning for Spatially Distributed Rainfall–Runoff

Modeling, EGU General Assembly 2024, Vienna, Austria, 14–19 Apr 2024, EGU24-8899, https://doi.org/10.5194/egusphere-egu24-8899, 2024.

Kratzert, F., Klotz, D., Shalev, G., Klambauer, G., Hochreiter, S., and Nearing, G.: Towards learning universal, regional, and local hydrological behaviors via machine learning applied to large-sample datasets, Hydrol. Earth Syst. Sci., 23, 5089–5110, https://doi.org/10.5194/hess-23-5089-2019, 2019b.

Kratzert, F., Gauch, M., Klotz, D., & Nearing, G. (2024). HESS Opinions: Never train a Long Short-Term Memory (LSTM) network on a single basin. *Hydrology and Earth System Sciences, 28*(17), 4187–4201. https://doi.org/10.5194/hess-28-4187-2024

Lees, T., Buechel, M., Anderson, B., Slater, L., Reece, S., Coxon, G., and Dadson, S. J.: Benchmarking data-driven rainfallrunoff models in Great Britain: a comparison of long shortterm memory (LSTM)-based models with four lumped conceptual models, Hydrol. Earth Syst. Sci., 25, 5517–5534, https://doi.org/10.5194/hess-25-5517-2021, 2021.

Loritz, R., Dolich, A., Acuña Espinoza, E., Ebeling, P., Guse, B., Götte, J., Hassler, S. K., Hauffe, C., Heidbüchel, I., Kiesel, J., Mälicke, M., Müller-Thomy, H., Stölzle, M., & Tarasova, L. (2024). CAMELS-DE: Hydro-meteorological time series and attributes for 1582 catchments in Germany. *Earth System Science Data, 16*(12), 5625–5642. https://doi.org/10.5194/essd-16-5625-2024

---

## Referee Comment (RC2)

**Review of "Beyond Observed Extremes: Can Hybrid Deep Learning Models Improve Flood Prediction?" by Guan et al. (2025)**

https://egusphere.copernicus.org/preprints/2025/egusphere-2025-1509/

Summary:

This study assesses the extrapolation skill of traditional conceptual hydrological models (HBV, GR4J and SIMHYD), a purely data-driven model (LSTM), and three hybrid versions in predicting unprecedented floods on a large-sample data set from Austria. The findings support current issues identified by the community: LSTM-based models outperform traditional models in in-sample interpolation, improve somewhat in extrapolation, but show extrapolation limits. Many questions and hypotheses about extrapolation skill are raised, but left uninvestigated and unanswered.

Overall evaluation:

The research questions raised in this paper definitely are within the scope of HESS. Unfortunately, these questions are mainly left unanswered. The study brings little additional insight to what has been proposed in the literature so far. Given some shortcomings in the methodology and a too shallow discussion, I recommend a rejection at this point and encourage the authors to follow up on the questions they have raised in this manuscript – they are relevant and point in the right direction; answers will turn this effort into a meaningful contribution.

Specific comments:

- Abstract: Large parts of the abstract read like a wishlist for the future; I would rather like to understand from the abstract what is achieved within the paper, and read about future work in the Conclusions/Discussion/Outlook section.

- L. 36: It comes as a bit of a surprise to me that data-driven models should be able to solve the extrapolation problem, since they purely rely on provided data and are said to lack extrapolation/generalization skill (although they have shown great skill in specific settings) – one or two additional sentences might help explaining why specifically data-driven models are expected to suffer less from "training bias" than physics-based models (I assume the point here is to allow for more flexibility in modeling as compared to rigid process-based structures that might be too much of a constraint?).

- L. 65: The paragraph on the actual contribution of this paper seems too short. First, it needs to be outlined which broad types of hybrid modeling approaches exist (post-processing of model outputs, time-dependent parameters, data-driven sub-components, …) and which ones the authors assume here (how are the physics-based models and the LSTM "integrated"?). Second, how do the authors intend to analyze unprecedented flood events? Will this be a synthetic analysis with synthetic forcing data? Or will certain events be excluded from

training to be used for testing? The paper lacks a clear formulation of the specific research question to be addressed.

- L. 80: "The historical observation period spans from 1981 to 2017 for most catchments, providing a comprehensive dataset for model evaluation." I assume that not the whole period was actually used for evaluation, but there was a split into training, validation, and testing?

- L. 125: "Given that we focus on simulating extremes, where input data are highly skewed, Min-Max scaling was used for feature preprocessing instead of standardization to better preserve extreme values." – This needs more attention, please explain in more detail what the min-max scaling does and whether, in principle, it would allow for predicting something outside of the training data range (concerning theoretical limits of extrapolation skill of LSTMs). Current research on these aspects exists (e.g., Baste et al., 2025), such that the literature review needs to be extended in that regard.

- Section 3.2: Does the LSTM predict sequence-to-1, or sequence-to-sequence?

- L. 136: "…thus generating intermediate state variables that reflect key physical processes that drive floods (see Table A1 for a summary of the intermediate state variables)" – since this is the only information provided about what makes the hybrid models hybrid, these variables need to be listed in the main body of the manuscript. Also, it needs to be discussed that the three conceptual models provide different intermediate variables, and hence the number and types of inputs differ between the three hybrid versions.

- L. 167: Why did the authors choose different loss functions for the conceptual hydrological models and the LSTM?

- Section 4.1: The NSE values achieved by the three conceptual models and also the pure LSTM are relatively low – can the authors provide an explanation? What is hard about modeling these catchments?

- L. 195: The pure LSTM does not outperform GR4J in validation as measured by NSE; only very slightly when looking at RMSE.

- L. 200: It seems all three hybrid models perform practically equally well, which triggers the question whether the provided inputs are actually informative for the LSTM, or whether it simply benefits from additional input channels that offer more flexibility. A test with randomized/nonsense input could provide some insight (idea presented by Bárdossy et al. at EGU25).

- L. 210: "RMSE values for out-of-sample floods were consistently higher across all models, likely due to the higher discharge magnitudes in these events, which lead to larger absolute simulation errors despite similar NSE values." – the absolute value of simulation errors enters RMSE and NSE in the same way (squared errors); if only the larger discrepancies for out-of-sample flood events were causing the deviations, the RMSE should actually show more similar results between in- and out-of-sample, because the root is reducing this effect. What about the normalization in the NSE? Did the authors normalize with the variance of the observations in-sample vs. out-of-sample? Then the in-samplevariance would be much smaller than the out-of-sample variance, which would explain that NSE stays relatively constant during both periods.

- L. 221: "The underestimation of in-sample flood peaks may be partly due to issues with the quality of rainfall and discharge data or catchment-specific characteristics not fully captured by the models." – the pure LSTM and also the hybrid versions do not care about the quality of rainfall so much, they can learn to bias-correct. I would rather argue that conceptual models are limited by volume balance closure (so yes, they struggle if rainfall or discharge data is imperfect), while data-driven models are limited by theoretical extrapolation limits (the way the activation functions are defined is designed for interpolation, so generally good skill in interpolation, but not suitable for extrapolation; again, see e.g. Baste et al., 2025).

- Section 4.4: The discussion raises many questions, but fails to answer at least some of them. I agree that, apparently, the different conceptual hydrological models leave different amounts of room for improvement (this is an almost trivial conclusion given that they show differences in performance); however, the fact that all three hybrid versions practically show the same performance in my view clearly triggers the questions whether (1) it is not the modeled information that helps the LSTM, but the added degrees of freedom, and (2) whether this shared performance shows us the performance limit given those degrees of freedom.

- L. 272: "We suggest synthetic design storms (Macdonald et al., 2025) can be generated carefully to demonstrate how hidden states in LSTMs saturate near the activation function bounds, capping the model's capacity to represent unprecedented events. Specifically, we can simulate a series of increasing storm intensities beyond the training range, evaluate the model's response to these inputs by tracking activation values in hidden states, and examine whether LSTM predictions plateau due to activation function saturation" – Yes, this is the type of analysis I would suggest to answer these questions (and which is currently pursued in the community, see, e.g., Baste et al., 2025, or Martel et al, 2024, both in the same journal). Why did the authors not try something in this direction in this case study? To me, the findings presented here are not new, surprising or enlightening; rather, they raise questions that are already being discussed by the hydrological community.

- L. 281: "Our current hybridization strategy, which integrates hydrological states, lays a foundation for such advancements…" – This statement is too strong. This type of "integration" has been proposed before (Frame et al., 2021) and used many times (e.g., Martel et al., 2024), and it is a very limited type of integration, given that model-predicted intermediate state variables are passed to the LSTM as inputs, which it can basically choose to use or ignore, whatever the data say. Several authors have pointed out that this post-processing type of hybridization is a very weak constraint for an LSTM (e.g., Frame et al., 2021; Álvarez Chaves et al., 2025). So neither is the proposed strategy new, nor will it be a basis for actual integration of physics into machine learning.

- L. 286: "In this study, models were implemented independently for each test catchment, without considering the potential benefits of a regional approach." – only now I learn that the LSTM models weren't trained over all catchments. It

has been discussed broadly in the community that LSTMs perform much worse when trained on individual basins (Kratzert et al., 2024), and the open question is whether this is only due to the vast amount of data available, or because the LSTM needs the diversity in the value ranges for extrapolation skill (e.g., Frame et al., 2022; Acuna et al., 2025). Training the LSTM on individual basins without any additional tweak to improve extrapolation skill doesn't provide an advance for the community.

References:

- Acuña Espinoza, E., Loritz, R., Kratzert, F., Klotz, D., Gauch, M., Álvarez Chaves, M., and Ehret, U. (2025): Analyzing the generalization capabilities of a hybrid hydrological model for extrapolation to extreme events, Hydrol. Earth Syst. Sci., 29, 1277–1294, https://doi.org/10.5194/hess-29-1277-2025.

- Álvarez Chaves, M., Acuña Espinoza, E., Ehret, U., and Guthke, A. (2025): When physics gets in the way: an entropy-based evaluation of conceptual constraints in hybrid hydrological models, EGUsphere [preprint], https://doi.org/10.5194/egusphere-2025-1699.

- Bardossy, A., Seidel, J., and Acuna, E. (2025): Is Artificial Intelligence the Ultimate Solution for Hydrological Modelling?, EGU General Assembly 2025, Vienna, Austria, 27 Apr–2 May 2025, EGU25-12083, https://doi.org/10.5194/egusphere-egu25-12083.

- Baste, S., Klotz, D., Espinoza, E. A., Bardossy, A., and Loritz, R. (2025): Unveiling the Limits of Deep Learning Models in Hydrological Extrapolation Tasks, EGUsphere [preprint], https://doi.org/10.5194/egusphere-2025-425.

- Frame, J.M., F. Kratzert, A. Raney II, M. Rahman, F.R. Salas, and G.S. Nearing (2021): "Post-Post-Processing the National Water Model with Long Short-Term Memory Networks for Streamflow Predictions and Model Diagnostics." Journal of the American Water Resources Association 57(6): 885–905. https://doi.org/10.1111/1752-1688.12964.

- Frame, J. M., Kratzert, F., Klotz, D., Gauch, M., Shalev, G., Gilon, O., Qualls, L. M., Gupta, H. V., and Nearing, G. S. (2022): Deep learning rainfall–runoff predictions of extreme events, Hydrol. Earth Syst. Sci., 26, 3377–3392, https://doi.org/10.5194/hess-26-3377-2022.

- Kratzert, F., Gauch, M., Klotz, D., and Nearing, G. (2024): HESS Opinions: Never train a Long Short-Term Memory (LSTM) network on a single basin, Hydrol. Earth Syst. Sci., 28, 4187–4201, https://doi.org/10.5194/hess-28-4187-2024.

- Martel, J.-L., Arsenault, R., Turcotte, R., Castañeda-Gonzalez, M., Brissette, F., Armstrong, W., Mailhot, E., Pelletier-Dumont, J., Lachance-Cloutier, S., Rondeau-Genesse, G., and Caron, L.-P. (2024): Exploring the ability of LSTM-based hydrological models to simulate streamflow time series for flood frequency analysis, EGUsphere [preprint], https://doi.org/10.5194/egusphere-2024-2134.